# Thermal Effects of Topical Hyperbaric Oxygen Therapy in Hard-to-Heal Wounds—A Pilot Study

**DOI:** 10.3390/ijerph18136737

**Published:** 2021-06-23

**Authors:** Teresa Kasprzyk-Kucewicz, Armand Cholewka, Beata Englisz-Jurgielewicz, Romualda Mucha, Michał Relich, Marek Kawecki, Karolina Sieroń, Patrycja Onak, Agata Stanek

**Affiliations:** 1Faculty of Science and Technology, University of Silesia, 40-007 Katowice, Poland; armand.cholewka@us.edu.pl (A.C.); beatkaenglish@gmail.com (B.E.-J.); relich.mic@gmail.com (M.R.); 2Department and Clinic of Internal Medicine, Angiology and Physical Medicine, Specialistic Hospital No. 2 in Bytom, Batorego 15 St., 41-902 Bytom, Poland; romam28@wp.pl; 3Department of Health Sciences, Technical-Humanistic Academy, 43-300 Bielsko-Biała, Poland; kpirm@ath.bielsko.pl; 4Department of Physical Medicine, School of Health Sciences in Katowice, Medical University of Silesia, Medyków Street 12, 40-752 Katowice, Poland; ksieron@hot.pl; 5Med Holding S.A. Specialist Hospital Named after prof. E. Michałowski, Strzelecka 9 St., 40-073 Katowice, Poland; patrycja.onak@medholding.eu; 6Department and Clinic of Internal Medicine, Angiology and Physical Medicine, Faculty of Medical Sciences in Zabrze, Medical University of Silesia, Batorego 15 St., 41-902 Bytom, Poland; astanek@tlen.pl

**Keywords:** thermal imaging, whole-body hyperbaric oxygen therapy, topical hyperbaric oxygen therapy, hard-to-heal wounds

## Abstract

Clinical studies have been performed to evaluate the thermal response of topical hyperbaric oxygen therapy (THBOT) in patients suffering from hard-to-heal wounds diagnosed as venous leg ulcers located on their lower extremities. It was found that this therapy leads to a temperature decrease in areas around the wound. Moreover, a minor temperature differentiation between all areas was seen in the third period of topical hyperbaric oxygen therapy (THBOT) that may suggest that microcirculation and thermoregulation improvement start the healing process. On the other hand, the results of the conducted studies seem to prove that thermal imaging may provide a safe and effective method of analyzing wound healing of hard-to-heal wounds being treated with THBOT. This is the first study that tries to show the possibilities of a very new method by evaluating treatment of hard-to-heal wounds using thermal imaging, similar to the hyperbaric oxygen therapy effects evaluated by thermal imaging and described previously. However, the first clinical results showed a decrease in temperature due to the THBOT session and some qualitative similarities in the decrease in temperature differentiation between the studied areas and the temperature effects obtained due to hyperbaric oxygen therapy.

## 1. Introduction

The use of thermal imaging in different medical branches has shown that body surface temperature brings valuable information in diagnostics in many medical branches as well as in the evaluation of treatment effects. It should be noted that body surface temperature is strictly associated with the tissues’ blood supply and metabolic activity. That is why a body temperature map brings important information in diagnostics, especially skin and superficial tissues [1,2].

The vertical posture of the human body causes higher blood pressure in the lower limbs due to the hydrostatic pressure resulting from the height difference between the heart and lower limbs. It is likely to be an aggravating factor in patients who suffer from circulatory failure and/or vascular and heart disease [3,4] and who can also suffer from chronic hard-to-heal wounds, whose time to heal may last from a few months to several years [5,6]. This vascular disease has a high prevalence among the population and significantly impacts the quality of life [5]. That is why early diagnosis, primarily non-invasive and relatively cheap, is the key to proper and fast treatment, and thermal imaging seems to be an easily performed diagnostic method required in different clinical practices that may be suitable for this purpose [7,8,9,10,11,12,13].

One of the most advanced methods in hard-to-heal wound treatment is hyperbaric oxygen therapy (HBOT). This method uses atmospheric-pressure air, which is 1.5 to 3 times higher than normal, that consists of 100% oxygen on the whole human body in a special compartment called a hyperbaric chamber or in a so-called monoplace where only one patient is provided therapy in a pure oxygen environment. During the therapy, patients sit inside the chamber with the higher-than-normal air pressure, simultaneously breathing pure oxygen through a mask or helmet. During HBOT, the patient is subjected to an oxygen pressure between 1.5 and 3 ATA (atmosphere absolute). One ATA corresponds to the average atmospheric pressure exerted at sea level, or 14.7 psi. In the case of a monoplace chamber, the patient is immersed in a nearly pure oxygen atmosphere without any helmet or mask. HBOT is widely applied to chronic wounds in which high oxygen pressure accelerates neovascularization, reducing swelling and inflammation. That leads to improved tissue perfusion, reduced edema, improved blood circulation, and thermoregulation in the area of the ulcer. As a result, the above-mentioned mechanisms of this method start the healing processes, as seen on thermal maps of wounds’ surroundings [14,15,16]. Moreover, the partial pressure of oxygen in alveolar air increases. The vesiculo-capillary gradient in the lungs increases and, most importantly in HBOT, the solubility of oxygen in blood plasma increases [14,17]. The main problem in HBOT seems to be connected with the cost of the hyperbaric chamber, so there are limited possibilities for using it for many patients that need such therapy. That is why a new, similar and noble hard-to-heal wound treatment technique is being developed, called topical hyperbaric oxygen therapy (THBOT). The apparatus can provide this type of treatment under pressure barely exceeding 1 ATA applied in a small chamber (called a tube) designed for single-limb therapy [17].

The processes occurring in topical hyperbaric oxygen therapy are similar to those in the whole-body treatment, causing the intensification of oxygen diffusion and increased oxygen partial pressure in the tissues due to hydration [18,19]. Previous research has shown HBOT thermal effects in hard-to-heal wounds, where the treatment cycles have caused a decreasing temperature in the regions of interest. Moreover, the isotherm area of crura was also decreasing in that paper, which suggests the healing processes occurred [13]. In contrast to HBOT, the mechanisms of THBOT’s actions are not yet fully known. That is why the presented work aimed to evaluate the thermal response of this very new therapy technique.

Even though the oxygen pressure is much lower in THBOT than in HBOT, the main goal of the presented work was to find if, and eventually how, such a new type of treatment will impact the thermal tissue response. The expected effects of THBOT on tissues seem to be the improvement of healing processes and, consequently, the decrease in the wound area and a temperature drop, which may bring some important information about the therapy effects for physicians.

THBOT is a relatively new and young therapy. However, it seems justified to perform research, which can show the usefulness of this method in hard-to-heal wound treatment. The treatments for chronic intractable ulcers of the lower limbs are not fully effective nowadays. Every new technique that can give positive treatment results should be considered in research studies, especially as only few studies have assessed how temperature therapy affects the evaluation of hyperbaric oxygen therapy. However, THBOT is much cheaper, easier to use, available to patients, and based on the same physical phenomenon. That is why the authors tried to check if, considering the nearly normal pressure of oxygen in the apparatus described for legs, only the thermal effects may be in agreement with those obtained with HBOT.

On the other hand, it is known that the healing processes of wounds are connected with tissue metabolism changes. The connection between temperature and metabolism allows thermal imaging to assess the healing processes of wounds [7,20].

## 2. Materials and Methods

### 2.1. Subjects

The study involved a group of patients who had hard-to-heal wounds, without positive effects of previous treatment using compression stockings and colloid dressing, treated with THBOT (THBOT group). All patients had chronic venous leg ulcers localized to their lower extremities caused by insufficiency of superficial veins. Chronic venous ulcers are defined as wounds that do not heal over 8 weeks despite optimal local treatment.

Before THBOT, each patient underwent a Doppler examination, performed to estimate the ulceration etiology. The exclusion criteria included the following: non-venous etiology of the ulcers; patients with ABI (ankle-brachial) index below 0.9; and other states in which physical therapy is contraindicated, such as acute infections, pregnancy, cancer disease, or acute deep thrombosis.

The research protocols were performed in compliance with the Declaration of Helsinki and were approved by the Bioethical Committee (permission no. KNW/0022/KB1/102/16). All subjects gave written consent for inclusion in the research.

The group consisted of 12 patients (8 women and 4 men) aged between 52 and 85.

### 2.2. Topical Hyperbaric Oxygen Therapy Procedures

All patients were treated in the Department and Clinic of Internal Diseases, Angiology and Physical Medicine in Bytom, Poland.

The patients were subjected to 10 therapy sessions of treatment that lasted 30 min each day excluding weekends. During the procedure, each patient was in a reclining position. At the same time, the affected limb was placed directly in the cylinder chamber of the apparatus, to which special tubes provided 100% oxygen from a bottle with a constant pressure of 1.5 ATA [21]. This procedure is included as the standard protocol of treatment. Patients qualified for therapy according to the guidelines of the Polish Wound Treatment Society [17]. Before therapy, all patients were informed about the safety conditions in the topical hyperbaric chamber using the OXYBARIA–S apparatus (FASER, Tarnowskie Góry, Poland) for extremities, which is shown in Figure 1.

THBOT was used as an adjunct to standard therapy of venous ulcers, which was the same for all patients. After completing each THBOT treatment, a hydrocolloid dressing Granuflex (ConvaTec, Reading, Berkshire, UK) was applied to the venous ulcer. Compression therapy with the use of an elastic stocking Ulcer X (Sigvaris, Switzerland) was applied in all examined patients. During the treatment, all patients were under the care of a specialist in vascular medicine (angiologist).

The whole treatment process was divided into three stages: the 1st, 5th, and 10th sessions of THBOT were considered as periods I, II, and III, respectively.

### 2.3. Thermal Imaging Procedure

Using an E60 Flir Systems thermal camera (FLIR Systems, Wilsonville, Oregon, United States), thermograms were obtained before and immediately after the THBOT session. The thermal imaging protocol consisted of three stages of imaging, I, II, and III, which involved imaging after the 1st, 5th, and 10th sessions, respectively. The study’s methodology was performed at the same distance of 1 m perpendicularly to the body following the Glamorgan Protocol [20,22]. Measurements were made in a particular room where the temperature was kept at 23.5 ± 1.0 °C with a humidity of 50%.

The areas taken into consideration for thermal analysis are presented in Figure 2a–e. The following areas were included:AR01—above the wound (from the middle of the wound until half the length of the tibia);AR02—wound marked according to physician’s diagnosis;AR03—below the wound.

To compare the inflammatory area changes in both therapies, it was necessary to introduce an isotherm threshold, set for every patient as the lower limb’s average temperature. As an isotherm threshold, the temperature level was set. All temperatures below the value of the isotherm threshold were cut off from the thermal map. The inflammatory area changes were analyzed as the ratio of the number of pixels counted from the 2nd and 3rd periods to that from the 1st. To assure the same dimensions of the considered pixels, the thermal images were always taken at the same distance of 1 m perpendicularly to the examined body. Figure 3 presents the isotherm area for the representative patient crura. The average limb temperature was calculated based on the entire limb area, and the isotherm threshold was set. The entire limb area coincided with the three ROIs mentioned above (AR01, AR02, and AR03). The change in the obtained isotherm area was used to calculate the surface inflammatory-state area, shown in Table 1 The total number of pixels inside the area was counted automatically by ThermaCam ResearcherPro 2.8 software.

### 2.4. Statistical Analysis

Statistical analysis was performed using Statistica 13.1 (StatSoft Polska, Kraków, Poland). Data are expressed as mean values ± standard deviation (SD). In the case of normal distribution and homogeneity of variance, Student’s t-tests were used. Furthermore, Wilcoxon tests were performed. A deeper analysis was performed by using Friedman’s ANOVA test. The significance level was set as *p* < 0.05. All significant results are marked on graphs.

## 3. Results

Figure 2 shows the thermal images obtained before and immediately after THBOT of a representative patient from the studied group suffering from a hard-to-heal wound, taken in the 1st and 3rd periods of treatment.

In addition, in the thermal analysis, the inflammatory areas were calculated using the isotherm threshold, set as the mean temperature of crura according to previous proposals in the literature [7,11,13]. Changes in the area were analyzed as the ratio of pixel numbers obtained in the 2nd and 3rd periods to the pixel number counted at the start of therapy—1st period (Table 2). The obtained results showed quite different thermal behavior between the two therapies observed, especially in the 2nd period.

The area percentage changes counted by thermal imaging during the three THBOT treatment periods obtained for all studied patients are presented in Table 1.

The obtained thermal images presented in Figure 2 show the mean temperature values (Table 2) and prove that the skin surface temperature decreased due to THBOT, which seems to be similar to the effect obtained in the literature [13]. Based on the data shown in Table 1, one can see the decrease in inflammatory state with time, which is presented in Figure 4.

It can clearly be seen that the inflammatory area characterized with a higher-than-average crura temperature (defined as isotherm area) decreased significantly with the increasing number of therapy sessions.

A deeper analysis of THBOT’s influence on thermal skin response is presented in Figure 5.

One can see from the thermal images and Figure 5 that the mean values of the pre- and post-session treatment temperatures values decreased. The only region of interest where the mean temperature difference increased was A2 during the first period of treatment; however, it was the wet area of the wound at the beginning of the treatment, which is usually seen in hard-to-heal wounds. The mean temperature differences above and below the wound showed a decreasing temperature tendency that may be explained by reducing edema, improving blood circulation, and thermoregulation in the ulcer surrounding [14,15,16].

The next step in data analysis was to take into consideration the mean temperature changes at AR01 and AR03 during all three periods of THBOT, which is presented in Figure 6.

The data shown in Figure 6 were intended to demonstrate THBOT’s influence on the temperature range obtained from the chosen region of interest—it is the parameter called ∆*A_i_/P_i_*, which is defined as the equation below:$$∆A_{i}/P_{i}={\left(T_{max}-T_{min}\right)}_{before\;THBOT}-{\left(T_{max}-T_{min}\right)}_{after\;THBOT}$$
where *A_i_* is the area from which the temperature is calculated, and *i* = 1, 2, 3; *P_i_*—the period of thermal imaging, defined as the 1st, 2nd, or 3rd period; *T_max_*—maximal temperature obtained in a given area; *T_min_*—minimal temperature obtained in a given area.

Such comparisons stem from the differentiation of surface areas because Areas 1 and 3 are usually dry in reverse to a wound, and a physician uses oximetry to assess the partial oxygen pressure values above and below the wound. However, the Friedman ANOVA test did not show statistically significant differences except for in Area 3 in THBOT, and a tendency of decreasing temperature range may be seen. However, for Area 3 in the 3rd period, statistically significant widening of the temperature range was obtained.

The obtained thermal parameters’ changes are complementary to the results presented in Figure 5, and such a drop in temperature in Area 3 at the end of treatment seems to be the proper treatment effect, which may be also in agreement with data in the literature [14,15,16,23].

We should consider that THBOT is a very new treatment method, and the tubes have just been introduced to hospitals, so the number of patients is still growing and the technique is developing. Moreover, the therapy may be provided for longer than 10 days.

## 4. Discussion

The aim of this study was to show the usefulness of thermal imaging in evaluating chronic wound treatment by use of THBOT. As the first study of THBOT’s thermal influence on leg ulcers, the research was based on a previous study that provides some promising results of HBOT’s healing effects in hard-to-heal wounds [13]. However, it should be pointed out that the conclusion about metabolism is indirectly due to the thermal images that show the body surface temperature.

Thermal imaging of the research group was provided before and after the 1st, 5th, and 10th sessions of healing. The whole area of the lower leg with a wound was imaged and divided into three areas (above and below the wound and the wound area), which were taken for analysis.

The main findings showed the surface inflammatory-state area and temperature changes of the mentioned leg areas in three treatment periods, which correspond with the previous work about HBOT temperature assessment. The findings agree with the previous work [13].

These findings may have implications for easy, fast, and non-invasive evaluation of treatment effects from the use of THBOT in hard-to-heal wound treatment. The quantitative data presented as a thermal map as well as the proposed thermal parameters may bring some new and valuable information for physicians, showing temperature changes around the wound that may be seen indirectly as metabolism; therefore, the healing process changes.

Analyzing the data shown in Table 1 and Figure 2, one can see that the counted isotherm area decreased during therapy. This may suggest a decrease in inflammatory-state areas. Moreover, the thermal images and figures show that the inflammation area changes in the two first treatment periods are more dynamic in their course. The drop in temperature measured before and after the THBOT sessions for the chosen areas can be seen in the thermal images shown in Figure 2 and the data in Table 2. Such an observation may be correlated with thermoregulation improvement after THBOT due to capillaries rebuilding and blood supply improvement, which is a desirable effect in the healing process. Such effects may be evaluated by mean temperature changes as well as the range of isotherm area that has been shown in the presented work. It seems to be entirely qualitative, similar to the effects obtained in thermal imaging evaluation of hard-to-heal wound treatment using hyperbaric oxygen therapy performed in chambers, as presented in previous papers [7,11,13].

The result is in agreement with the areas representing inflammatory-state borders and their changes during therapy, as presented in Figure 4. Such observations may also be in agreement with periods of treatment of hard-to-heal wounds shown in the literature, where the middle period is tissue granulation when metabolism speeds up due to neoangiogenesis and capillaries appear on the surface of a wound that influences the degree of inflammation and surface temperature derived, as seen in the thermal images. Moreover, the obtained results suggest the initiation of processes related to a reduction in edema and inflammation and improvement in the wound’s oxygenation [13,14,24,25,26]. It should also be noted that these are pilot studies, and once the COVID-19 pandemic ends, further studies will be provided to see the similarities and differences between HBOT and THBOT. It may be a very interesting comparison due to the large discrepancies in the apparatuses’ prices and the therapy availability for patients.

It should be underlined that THBOT led to a significant decrease in the inflammatory state at the end of the therapy. It can be indirectly seen as a temperature decrease due to each session of THBOT, confirmed by statistical analysis with *p* = 0.01. It seems to be entirely qualitative, similar to the effects obtained in thermal imaging evaluation of hard-to-heal wound treatment using hyperbaric oxygen therapy performed in chambers, as presented in previous papers [7,11]. This was the needed effect, and it was obtained with the treatment. It should be noted that THBOT uses a normal pressure of oxygen, but it additionally adds ozone into the body. Used in THBOT, ozone has a powerful disinfection and oxygenation action, leading to killing of bacteria, fungi, and viruses. It is considered the second line of therapy for refractory wounds [27]. Delivery of these gases to a wound is an essential advantage and leads to accelerating neovascularization and reducing swelling and inflammation of the affected area [13,14,15,16]. That is why during therapy, an improvement of perfusions and reduction in edema is observed. This is an essential part of healing that is seen in thermal images as a reduction in the areas characterized by a higher temperature due to improved blood circulation and thermoregulation in the ulcer area. The therapy’s mentioned advantages allow antibiotics to be delivered to the wound area, thus facilitating pharmaceutical therapy [14].

Moreover, the thermal parameters obtained from thermal images during one cycle of 10 THBOT sessions may prove the decrease in the mean temperature, which can be seen in Table 2 and Figure 5.

Additionally, healing processes leading to tissue reconstruction seem to be similar to those previously reported in the literature for whole-body hyperbaric oxygen therapy [13,19,28]. It seems that the observed temperature and dimensions of inflammatory changes may give additional diagnostic information for the physician because they are strictly connected with the range of metabolically active tissue and may suggest the beginning of the healing process—its growth or decrease. This may help the clinician to decide the next step in the treatment, especially as thermal imaging is an entirely non-invasive, simple, and fast technique [7,13]. Such evaluation may be called functional due to the relation between temperature, tissue metabolism, and thermoregulation and, thus, the changes in microcirculation and main blood supply.

Based on the data shown in Figure 5 and Figure 6, the temperature drop observed in the regions of interest seems to confirm an essential process occurring in hard-to-heal wounds treated with THBOT, which was mentioned before. Moreover, we may see an increasing tendency in the temperature rise of AR01 observed in the last treatment period in THBOT. Such thermal behavior of tissue may be correlated with the therapy’s properties but should be explored more deeply and confirmed by further studies.

Based on the obtained results, the impact of THBOT in the treatment of hard-to-heal wounds generally leads to a decrease in the temperature of the areas around the wound and the area of the wound itself, proportionally to the number of treatment sessions. Such observations may correlate with the improvement of thermoregulation and, in turn, the increased oxygen and ozone supply in the tissues. Moreover, the tendency to increase the temperature of the wound area in the middle period of treatment is clearly seen. However, this effect was reported as more extensive in the literature during HBOT [13]. The tendency to increase the temperature range in the 2nd period can also be seen. This may suggest that the metabolism response changes and correlates with angiogenesis during therapy using oxygen under raised pressure, which is also in agreement with healing processes [14,15,29,30,31].

It should be pointed out that other studies proved that topical oxygen therapy shortens the healing time in patients with nonhealing diabetic foot ulcers and may have the ability to encourage the growth of aerobic members of the wound microbiome and be an effective alternative to using antibiotics in patients with chronic diabetic foot ulcer, which also seems to correlate with the discussed results [30,31].

With this paper as a pilot study, further research related to THBOT is planned in the future. Further studies will be related to the expansion of the research group and the correlation between the biochemical and wound evaluation index.

## 5. Conclusions

The conducted studies have shown that the inflammatory area changes during THBOT. THBOT indirectly leads to a decrease in the temperature of the areas around the wound at a statistically significant level. The temperature drop observed in the regions of interest was the strongest in the 3rd period of treatment, which may be related to microcirculation as well as the improvement in thermoregulation, i.e., the healing process.

The obtained results seem to prove that thermal imaging may provide a safe and effective method of analyzing wound healing in hard-to-heal wounds being treated with THBOT. However, more studies, including clinical trials, are needed to analyze the mechanisms of this still brand-new therapy technique.

Moreover, THBOT is much cheaper, less complicated to use, and available to much more patients than HBOT is, which is frequently used nowadays, and the very first results of a thermal evaluation of this treatment method seem to be qualitatively similar in terms of the decrease in temperature differentiation between the studied areas and the temperature effects obtained from hyperbaric oxygen therapy, which is promising and requires further and more profound studies.

## Figures and Tables

**Figure 1 ijerph-18-06737-f001:**
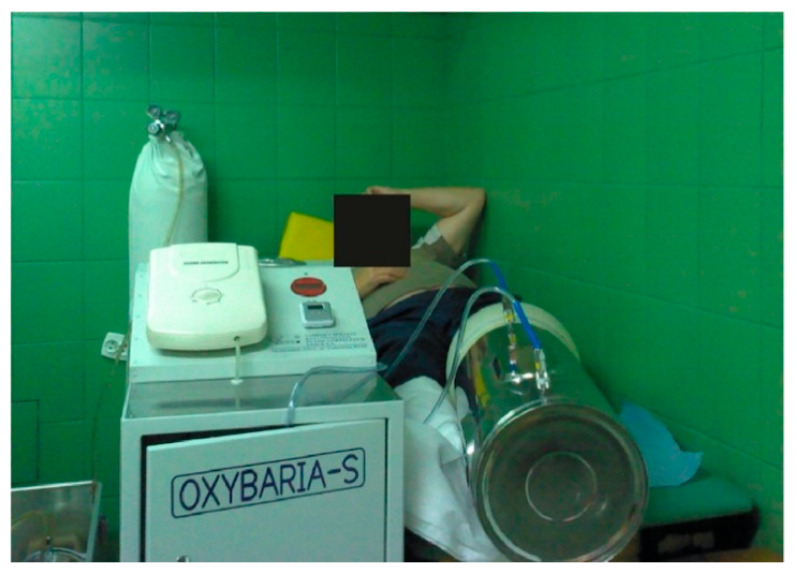
Topical hyperbaric chamber—the OXYBARIA—S apparatus (FASER, Tarnowskie Góry, Poland).

**Figure 2 ijerph-18-06737-f002:**
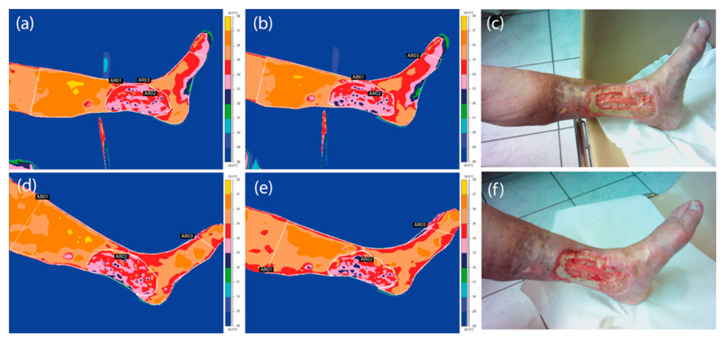
Thermal images of a representative patient suffering from a hard-to-heal wound taken in the 1st (**a**–**c**) and 3rd (**d**–**f**) period of treatment before (**a**,**d**) and after (**b**,**e**) THBOT with corresponding digital photos of the crura.

**Figure 3 ijerph-18-06737-f003:**
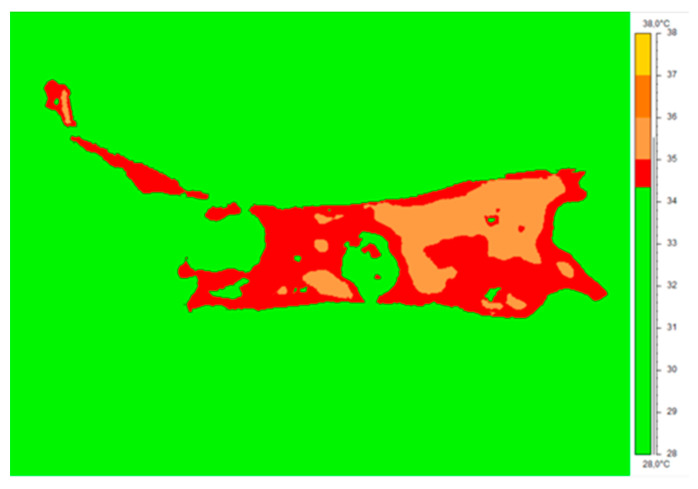
The isotherm area for a representative patient limb. The isotherm temperature was set as the mean temperature of the whole limb. All pixels with a temperature lower than the threshold (green color) were not counted in ROIs’ mean temperature values.

**Figure 4 ijerph-18-06737-f004:**
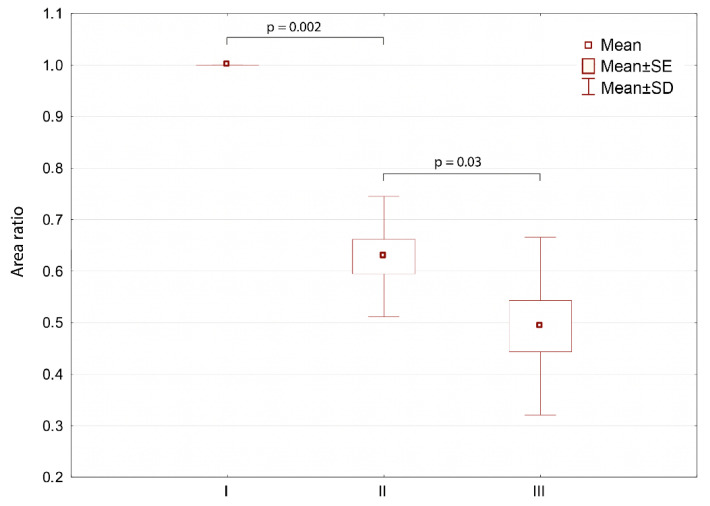
Isotherm area changes obtained during therapy as an average for all studied patients treated with THBOT, taken into account as a ratio of pixel numbers from the 2nd and 3rd periods to that in the 1st period.

**Figure 5 ijerph-18-06737-f005:**
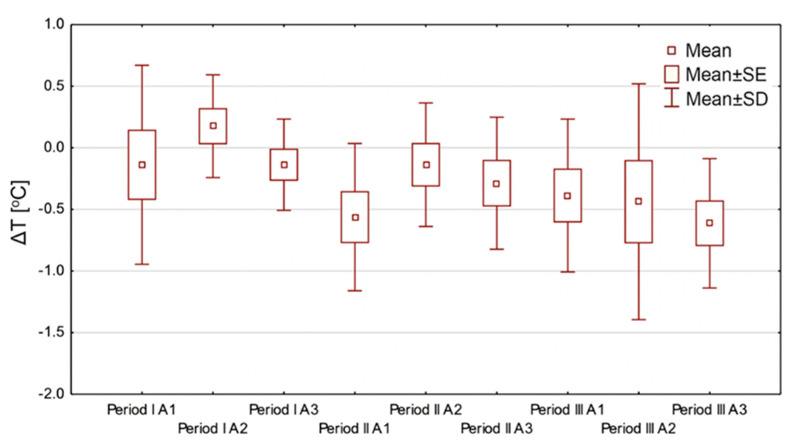
Summary graph showing differences in mean temperature values (SE—standard error; SD—standard deviation) for all studied areas before and after THBOT treatment, in all three treatment periods for all patients, where ∆T is defined as (T mean after THBOT-T mean before THBOT).

**Figure 6 ijerph-18-06737-f006:**
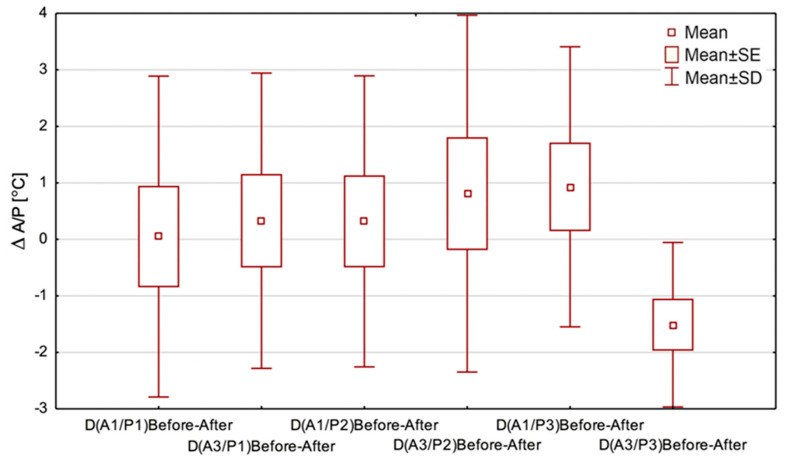
The temperature range changes in Area 1 and Area 3 during the three periods of THBOT, where, e.g., ∆(A1/P1)Before-After is defined as (Tmax-Tmin) before THBOT—(Tmax-Tmin) after THBOT in Area 1 and Period 1. SE—standard error; SD—standard deviation.

**Table 1 ijerph-18-06737-t001:** Area changes counted by thermal imaging during the three THBOT treatment periods obtained for all studied patients.

Surface Inflammatory State Area Changes (Ratio of Pixel Number Obtained in the 2nd and 3rd Periods to That in 1st Period)		
**I**	**II**	**III**
1.00	0.69	0.54
1.00	0.46	0.57
1.00	0.66	0.43
1.00	0.64	0.77
1.00	0.52	0.09
1.00	0.47	0.38
1.00	0.65	0.68
1.00	0.90	0.40
1.00	0.69	0.61
1.00	0.64	0.45
1.00	0.62	0.48
1.00	0.60	0.52

**Table 2 ijerph-18-06737-t002:** Average temperature of the studied areas obtained for the representative patient taken before and after THBOT during three periods of treatment.

Area	Period I	Period II	Period III
	T (°C) After THBOT		
AR01	36.1	33.9	34.4
AR02	36.2	34.0	34.5
AR03	35.7	34.0	35.1

## Data Availability

Data available on request due to restrictions eg privacy or ethical The data presented in this study are available on request from the corresponding author. The data are not publicly available due to ethical and privacy policy.

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
