# Peer review of "Thermal Effects of Topical Hyperbaric Oxygen Therapy in Hard-to-Heal Wounds—A Pilot Study"

_ijerph, 2021, doi:10.3390/ijerph18136737_

Round 1

Reviewer 1 Report

Interesting data but clinical relevance should be more clearly stated.

  1. What was the standard therapy and compression therapy for these patients during HBOT therapy, was it same for all?
  2. Was there correlation between wound healing times and the thermal parameters the authors present?
  3. In the discussion section, the clinocal relevance of the findings should be more discussed.

Reviewer 2 Report

The manuscript has not have reach merits. The number of participants is not significance, very short with 12 patients , and poor description of the methodology/statistics.  The text is structured and is dificult to follow. there are several language related errors and thus the manuscript requires a thorough checking by a native speaker of English.

2. Importance

The topic has not potential to offer more solid evidence based information about the potential of clinical outcome of Thermal Effects of Topical Hyperbaric Oxygen Therapy in Hard-to-Heal Wounds

Hypotheses are needed. What do the authors expect to find based on previous work? This would also be a good place to justify why this study is needed.3. Justification/Rationale

The justification of this study could be strengthened by explaining what this study adds to the pre-existing literature. One very important finding of the study should appear already in Abstract or in Conclusions: This information is not conclusive related with “the results of conducted studies seem to prove that thermal imaging may provide a safe and effective method of analyzing wound healing of hard-to-heal wounds being treated with THBOT” and no clear evidence that the authors propose a experiment must have been conducted rigorously.

Please make it clearer already in the Introduction, what new this study has to offer (i.e. in what way it differs from earlier studies).

3. Methods/Approach

The methodology is not rationale, seriously flawed from the methodological point of view related type of the design and trial registry before the first participant is enrolled into the study creates a publicly-available record of the researcher’s intentions; including key details such as how many patients they need to recruit, what their outcomes will be, how they intend to measure these outcomes, etc.

The tool used used are standardized and not well suited for the purpose. How was assessed the calibration of the both of them  systems? A external process of calibration is needed in a high standard level of pressures measurements. If the authors wants to “validate” this clinical-marketing tool for a research propose. This device don’t have this process, so it´s automatically discarded to a good research propose.
I appreciate that the authors included a sample size calculation

4. Results/Findings

These issues invalidates completely all the trial.
5. Discussion

Discussion is muddled, confusing to follow and repeats somewhat the Introduction. Furthermore, some more shortcomings should be included, particularly the fact the bias found. The authors could also consider some more ideas for further studies.

6. Conclusions

Write this section part again, clearly and include causative conclusions are warranted.

7. Figures.

please reconsider remove the figure one and two are not informative.

Reviewer 3 Report

I am very happy to review your valuable manuscript. This study seemed to investigate the effect of Topical hyperbaric oxygen therapy (THBOT) on chronic intractable wounds. I felt that this is a positive one in that there was no effective treatment for chronic intractable ulcer at the lower extremities in the current state. However, I think that the authors should answer the following several concerns.

First, this study was a single group study without a control group. I recommend that the authors add the control group to clear the authors’ thesis. If the authors would like to publish this manuscript in the current form, I think that it is not appropriate in the form of the original article at the current level. The case series is just valid.

Second, since THBOT is not a popular and well-known treatment method, a clear and additional explanation should be needed to explain this treatment method to the readers. I recommend that adding pictures or additional figures of how the treatment actually was conducted. In addition, contrary to what the name or definition of THBOT, it seems that contact with oxygen may have an effect rather than pressure has a therapeutic effect. This needs to be explained.

Third, to prove the authors' thesis, it is judged that not only the effects of the thermal imaging machine, but also clinical biochemical indicators and wound evaluation index, etc. will be additionally needed. It is difficult to say that simply the drop in temperature around the wound caused the wound to be relieved.

Thank you.

Round 2

Reviewer 2 Report

The paper has much improved, and although I have reservations about the interpretation of the data, and the strength of evidence for the clinical message, I think the article presents the data well enough for readers to judge themselves. I would recommend publication.

Author Response

Thank you very much for this comment. The English language has been corrected by native speaker. We are grateful for paper recommendation.

Reviewer 3 Report

Thank you for your active revision. However, there have still remained to be further added to prove the authors' thesis. As I had commented in the first review, this study should have had the control group. Unless you are able to add the control group. Article format should be changed to case report or series. 

Author Response

This manuscript is a resubmission of an earlier submission. The following is a list of the peer review reports and author responses from that submission.

Round 1

Reviewer 1 Report

Thank you for considering the suggested changes.

Reviewer 2 Report

Dear authors
I read carefully the answers to my questions.
I believe that the article meets the quality criteria to be published